# Research Progress of Genomics Applications in Secondary Metabolites of Medicinal Plants: A Case Study in Safflower

**DOI:** 10.3390/ijms26083867

**Published:** 2025-04-19

**Authors:** Zhihua Wu, Yan Hu, Ruru Hao, Ruting Li, Xiaona Lu, Mdachi Winfrida Itale, Yang Yuan, Xiaoxian Zhu, Jiaqiang Zhang, Longxiang Wang, Meihao Sun, Xianfei Hou

**Affiliations:** 1College of Life Sciences, Zhejiang Normal University, Jinhua 321004, China; yanh_nikki@163.com (Y.H.); 18756879273@163.com (R.H.); 18672754906@163.com (R.L.); 13336350698@163.com (X.L.); weenieitale@gmail.com (M.W.I.); xx-zhu@zjnu.edu.cn (X.Z.); wang.longxiang@zjnu.edu.cn (L.W.); mhsun@zjnu.cn (M.S.); 2National Key Laboratory of Crop Genetic Improvement, Hubei Hongshan Laboratory, Huazhong Agricultural University, Wuhan 430070, China; yangyuan9109@163.com; 3Zhejiang Institute of Landscape Plants and Flowers, Hangzhou 310053, China; zhangqiang414@126.com; 4Crop Research Institute, Xinjiang Academy of Agricultural Sciences, Urumqi 830091, China

**Keywords:** medicinal plants, genomics, secondary metabolites, regulation, safflower

## Abstract

Medicinal plants, recognized as significant natural resources, have gained prominence in response to the increasing global demand for herbal medicines, necessitating the large-scale production of these plants and their derivatives. Medicinal plants are exposed to a variety of internal and external factors that interact to influence the biosynthesis and accumulation of secondary metabolites. With the rapid development of omics technologies such as genomics, transcriptomics, proteomics, and metabolomics, multi-omics technologies have become important tools for revealing the complexity and functionality of organisms. They are conducive to further uncovering the biological activities of secondary metabolites in medicinal plants and clarifying the molecular mechanisms underlying the production of secondary metabolites. Also, artificial intelligence (AI) technology accelerates the comprehensive utilization of high-dimensional datasets and offers transformative potential for multi-omics analysis. However, there is currently no systematic review summarizing the genomic mechanisms of secondary metabolite biosynthesis in medicinal plants. Safflower (*Carthamus tinctorius* L.) has rich and diverse bioactive flavonoids, among of which Hydroxysafflor yellow A (HSYA) is specific to safflower and emerging as a potential medication for treating a wide range of diseases. Hence, significant progress has been made in the study of safflower as an excellent example for the regulation of secondary metabolites in medicinal plants in recent years. Here, we review the progress on the understanding of the regulation of main secondary metabolites at the multi-omics level, and summarize the influence of various factors on their types and contents, with a particular focus on safflower flavonoids. This review aims to provide a comprehensive insight into the regulatory mechanisms of secondary metabolite biosynthesis from the perspective of genomics.

## 1. Introduction

The advancement of science and technology has heightened global awareness of the significance of medicinal plants. According to data from the World Health Organization, approximately 80% of the global population relies on natural medicines. The exploration and utilization of genetic resources of medicinal plants provide a solid foundation for the traditional Chinese medicine industry. The development of high-throughput sequencing technology can elucidate the biosynthetic pathways of plant secondary metabolites at the genomic level, and mine key enzymes associated with their bioactive products, which will allow the application of synthetic biology in the in vitro synthesis of natural products in medical plants themselves or microorganisms to flourish. While certain medicinal plants, such as *Panax ginseng*, *Camptotheca acuminata*, and *Panax notoginseng* [1], have undergone complete whole-genome sequencing, the majority of medicinal plants still face challenges, including the ambiguity of their pharmacological active ingredients, the lack of biosynthetic mechanisms, and the difficulties of data mining at the genomic level. These obstacles impede the further development and utilization of genetic resources from medicinal plants. Ongoing advancements in genomics have accelerated the application of single-omics disciplines, including transcriptomics, proteomics, and metabolomics. In recent years, the rise of single-cell transcriptomics (scRNA-seq) and spatial transcriptomics technologies has provided unprecedented high-resolution insights into gene expression at the cellular level. Spatial transcriptomics addresses the limitations of traditional single-cell sequencing, which often loses critical spatial information about cells. This innovative technology enables the detection of gene expression while preserving the spatial position of cells, thereby elucidating the distribution and function of cells within tissues. However, single-omics methodologies are constrained in their ability to elucidate the global gene regulation under the frame of central dogma from genes to traits at the genomic level [2]. Consequently, the integrated application of these technologies—known as integrative multi-omics analysis—has emerged as a prominent trend in research of plant secondary metabolites. Furthermore, as a powerful tool for creativity, artificial intelligence (AI) shows great potential for application in the field of medicinal plants and plant-derived medicines based on large-scale genomic data. Using efficient algorithms of machine learning and deep learning (DL) in multi-omics data analysis [3], AI enhances accuracy in gene mining for precision medicine by reducing data dimensionality and unraveling the complexities of biological processes. With AI being more frequently used in multi-omics analysis, we expect a paradigm shift in gene mining and biosynthesis of plant secondary metabolites to be driven by AI-based multi-omics technologies.

The pharmacological properties of medicinal plants are closely associated with their rich and diverse secondary metabolites, also called natural products. Major plant secondary metabolites, including alkaloids, terpenoids, phenols, and quinones, exhibit a range of biological activities, such as antioxidant, antibacterial, anti-inflammatory, and anti-tumor effects. They constitute the primary active constituents in various traditional Chinese medicines as well as contemporary pharmaceuticals [4,5,6]. Therefore, secondary metabolites serve not only as the foundational elements that enable medicinal plants to retain their therapeutic efficacy but also as critical parameters for assessing the quality of medicinal plants. Nevertheless, the production of secondary metabolites in medicinal plants is often limited, and their biosynthetic pathways typically involve a series of enzymatic reactions and complex regulatory networks that require further investigation. In addition, environmental factors, including light, temperature, microorganisms, and the genetic characteristics of the plants themselves, significantly influence the synthesis and accumulation of secondary metabolites [7,8]. Multi-omics analysis provides a more comprehensive and systematic perspective for an in-depth understanding of this problem. For example, safflower (*Carthamus tinctorius* L.) is an underutilized economic herb with both medicinal and edible value, which belongs to the genus *Carthamus* within the Asteraceae family. It contains a wide variety of bioactive flavonoids, such as Hydroxysafflor yellow A (HSYA), safflor yellow B, anhydrosafflor yellow B (AHSYB), and kaempferol. These flavonoids exhibit significant medicinal properties and have extensive applications in the prevention of cardiovascular diseases, including arteriosclerosis, hypercholesterolemia, and hyperlipidemia, as well as anti-cancer, anti-oxidation, anti-apoptosis, and anti-allergy effects [9]. In recent years, due to the potential application prospects of safflower flavonoids, multi-omics technology has also been applied to the study of the biosynthesis mechanism of safflower flavonoids.

Here, a search utilizing the terms “medicinal plants” and “genomics” was performed across the CNKI, PubMed, and Web of Science databases. Over the past decade, there has been a consistent increase in research in the field of medicinal plants along with the development of genomics technology (Figure 1a). The years 2021 and 2022 recorded the highest volume of publications concerning the genomics of medicinal plants, with 1028 and 1064 papers, respectively. An analysis of the data from 2014 to 2024 reveals that the top three countries contributing to the body of research on medicinal plant genomics include China (4240 papers), the United States (1435 papers), and India (738 papers). Additionally, the United Kingdom, Germany, and South Korea have demonstrated significant research engagement in this field, collectively advancing the comprehensive development of global medicinal plant genomics research (Figure 1b). In conclusion, genomics research of medicinal plants has emerged as a prominent area of interest within contemporary biological, medical, and pharmaceutical research. This article aims to review the latest advancements and challenges in genomic technologies related to the secondary metabolites of medicinal plants, focusing on three primary research areas: omics technologies, secondary metabolites, and their environmental driving factors. Furthermore, it anticipates exploring the molecular mechanisms underlying the interactions between the secondary metabolites and their environment, as well as advancements in genetic engineering and breeding strategies, with the objective of fostering significant progress in the field of medicinal plant genomics.

## 2. The Combined Effect of Genetic Regulation and Environmental Factors on Secondary Metabolites

### 2.1. Composition and Function of Major Secondary Metabolites in Plants

According to their chemical structure and characteristics, plant secondary metabolites can be mainly classified into four categories: alkaloids, terpenoids, phenols, and quinones. These compounds not only serve to protect plants from herbivorous insects and pathogenic bacteria but also have a diverse array of pharmacological activities, including anti-tumor effects, analgesic properties, and treatment for cardiovascular and cerebrovascular diseases, as well as anti-malarial effects (Figure 2, Table 1). Consequently, their applications span across medicine and pharmacy, endowing them with significant economic value and profound social importance.

Alkaloids are widely present in the plant kingdom and are mainly distributed in higher plants, such as Ranunculaceae, Leguminosae, Papaveraceae, Menispermaceae, and Loganiaceae [59]. Alkaloids are a class of nitrogen-containing basic organic compounds, most of which have complex ring structures, with nitrogen often incorporated within the rings. Currently, global public health is facing significant challenges posed by cancer, which is the second most common cause of death in developing countries. Compared to traditional cancer treatment methods that are limited by insufficient effective delivery of drugs to tumor tissues, drug resistance, and damage to normal tissues, natural compounds with antioxidant properties have shown potential as new cancer treatment options. Alkaloids have a significant role in the field of cancer treatment [60]. Vincristine is an anti-tumor drug that targets microtubule depolymerization, exerting its anti-tumor effects by inducing the production of reactive oxygen species, which convert tumor-associated macrophages from the M2 phenotype to the pro-inflammatory M1 phenotype [10].

Terpenoids can be viewed as a class of natural compounds made from isoprene or isopentane linked in various ways. According to their structures, they can be divided into monoterpenes, diterpenes, sesquiterpenes, triterpenes, and polyterpenes, etc. Terpenoids are widely distributed and diverse, and are the most abundant class of plant secondary metabolite compounds. Terpenoids currently put into clinical use include the anti-malarial drug artemisinin, the anti-tumor drug paclitaxel, and the drug tanshinone for cardiovascular diseases. Artemisinin is a highly effective anti-malarial compound extracted from *Artemisia annua* L. It can activate and bind to proteins in *Plasmodium* to inactivate them [25].

The chemical structure of phenolic compounds is characterized by one or more hydroxyl groups attached to a benzene ring. These compounds can be categorized into three types based on their forms: free state, soluble bound state, and insoluble bound state. A study conducted a comprehensive analysis of free, glycosylated, esterified, and insoluble phenolic compounds in safflower seeds for the first time, identifying a total of 55 distinct phenolic compounds. The results indicated that most of these compounds existed in free form, while only a small fraction was found in insoluble form [61]. Flavonoids, as one of the most well-known and studied classes of phenolic compounds, have been isolated and characterized in more than 10,000 species to date [62]. In terms of chemical structure, flavonoids possess a basic skeleton consisting of three rings (C6-C3-C6). As a type of flavonoid specifically existing in safflower, HSYA is the main bioactive substance of safflower flower, which has the efficacy of activating blood circulation and improving blood stasis, and is of great value in the prevention and treatment of cardiovascular and cerebral vascular diseases [9]. HSYA has the capacity to safeguard the integrity of the blood–brain barrier in ischemic regions by suppressing the expression of pro-inflammatory genes and apoptosis-related genes through the NF-κB-related pathways [63].

Quinones are compounds characterized by an intramolecular unsaturated cyclic diketone structure or easily converted into this structure. In most instances, they serve as crucial electron transfer mediators in biological systems through reversible redox processes. Existing research indicates that quinones exhibit a variety of pharmacological activities, including anti-tumor, antibacterial, antifungal, antiviral, anti-Alzheimer’s, and anti-malarial effects [64]. Among them, anthraquinones exhibit a diverse array of pharmacological properties. In recent years, pharmaceutical chemists have been exploring the potential for modifying anthraquinone rings. They have found that the interaction between anthraquinone rings and biological targets involved in cancer progression can be enhanced by the incorporation of suitable functional groups, such as amino groups, hydroxyl groups, and their derivatives. The resulting anthraquinone derivatives exhibit improved anti-cancer characteristics and reduced toxicity, effectively addressing multidrug resistance [65]. Coenzyme Q10, a fat-soluble quinone compound, is commonly used as a supplement to promote cardiovascular health. Related research has investigated the molecular mechanisms involved in regulating the side chain length of plant coenzyme Q. By employing guided editing technology, researchers modified five amino acids in the Coq1 enzyme within the rice genome, resulting in the creation of a new rice germplasm capable of synthesizing coenzyme Q10 [56].

### 2.2. Genetic Regulation of Major Secondary Metabolites in Plants

The genetic regulation of plant secondary metabolites is a complex process involving multiple factors such as gene expression, transcription factors, and epigenetic modifications. These factors work together to precisely control the synthesis and accumulation of secondary metabolites. Among different plant species or even different varieties of the same plant, differences in gene sequences and related factors may result in alterations to key enzymes or regulatory elements within the synthesis pathways of secondary metabolites, thereby affecting the types, concentrations, and activities of secondary metabolites.

The primary types of plant alkaloids include terpenoid indole alkaloids (TIAs), benzylisoquinoline alkaloids (BIAs), hyoscyamine, nicotine, and purine alkaloids, among others. Each of these alkaloids follows specific biosynthetic pathways within plants. For instance, in the case of monoterpene indole alkaloids (MIAs), strictosidine synthase (STR) is one of the key enzymes in this pathway. The product, strictosidine, serves as an important intermediate that can be further transformed into various alkaloids, including vinblastine, quinine, and brucine [66]. At the level of transcription regulation, it has been demonstrated that the transcription factors ORCA2 and ORCA3 interact with jasmonate and the elicitor-responsive element on the STR promoter, leading to the synthesis of TIA. Additionally, ORCA4, ORCA5, and ORCA6 are also part of the ORCA cluster, which can further activate the STR promoter [67]. In addition, the purine alkaloid theacrine exhibits sedative and antidepressant effects, and its biosynthesis involves a key rate-limiting enzyme known as caffeine oxidase. The gene encoding caffeine oxidase (*CsCDH*) has been cloned, and the caffeine oxidase activity of CsCDH in tea plants was confirmed through transient expression in tobacco and antisense oligonucleotide interference experiments, which laid a foundation for clarifying the biosynthetic pathway of theacrine [68]. In plants, terpenoids can be synthesized through two primary pathways: the mevalonate (MVA) pathway and the 2-C-methyl-D-erythritol-4-phosphate (MEP) pathway. Isopentenyl diphosphate and dimethylallyl diphosphate form geranyl diphosphate, farnesyl diphosphate, and geranylgeranyl diphosphate under the action of isopentenyl transferase. These acyclic intermediates subsequently give rise to various terpenoids through the process of terpenoid synthesis (TPS) [69]. TPS is one of the key enzymes that initiates terpenoid biosynthesis, and its activity determines both the types and the synthesis efficiency of terpenoids. For example, HY5 in *Arabidopsis thaliana* can regulate the terpene synthase *AtTPS03*, thereby influencing terpene biosynthesis [70]. It was found that the ancestral *TPS* gene encoded bifunctional diterpene synthases I and II, which produce ent-kaurene, a compound essential for all existing lineages of land plants to synthesize plant hormones. The parallel evolution of diterpene synthases I and II also resulted in members of the TPS-e/f and TPS-c subfamilies contributing to secondary metabolism, which provided an evolutionary background for the *TPS* gene to synthesize a diverse array of natural terpenoids across various lineages of land plants [71]. In addition, one study elucidated the origin and evolutionary mechanisms of diterpenoid alkaloid synthesis genes in *Aconitum* plants through multi-group analysis. They identified key node genes within the co-expression weight network that are involved in the formation of diterpenoid precursors, the construction of diterpenoid alkaloid skeletons, and the post-modification of these skeletons, including enzymes encoding target-kaurene oxidases and aminotransferases. The study emphasized the key role of the *TPS* gene in plant secondary metabolism [72].

Flavonoids, a class of phenolic compounds, are among the most extensively studied. In the anabolic pathway of plant flavonoids, phenylalanine generates p-Coumaroyl CoA through the action of PAL, C4H, and 4CL, and then produces naringenin chalcone under the catalysis of CHS, and naringenin chalcone synthesizes downstream products under the action of different enzymes [9]. CHS is the first restriction enzyme in this pathway. Related studies have successfully cloned *CtCHS1* and demonstrated its involvement in the conversion of p-Coumaroyl CoA and malonyl CoA substrates in safflower to naringenin chalcone, which provides valuable genetic resources for the in vitro synthesis of flavonoids [73]. At the level of transcriptional regulation, the biosynthesis of flavonoids in plants is mainly coordinated by the MBW complex, which consists of MYB transcription factors, WD40, and bHLH proteins [67]. Overexpression of *CtbHLH41*, *CtMYB63*, or *CtWD40-6* in safflower can significantly enhance the content of HSYA [74]. Additionally, it was found that DcbHLH5 in *Dracaena cambodiana* can increase flavonoid content and enhance anti-UV-B capacity by activating *CHS1*, *CHS2*, and *CHI1* [75]. Furthermore, non-coding RNA plays a crucial role in flavonoid synthesis. miRNA can regulate the formation of flavonoids by acting on structural genes or indirectly utilizing the MBW transcription complex containing MYB-bHLH-WD40 [76]; two long non-coding RNAs (lncRNAs), LNC1 and LNC2, can act as endogenous target mimics of miR156a and miR828a, respectively, reducing the expression of *SPL9* and inducing the expression of *MYB114*, resulting in an increase and a decrease in anthocyanin content [77].

In vivo synthesis of anthraquinone is a complex process that involves numerous metabolites derived from various metabolic pathways. Currently, the biosynthetic pathway of anthraquinone remains incompletely understood. Generally, there are two primary pathways for anthraquinone biosynthesis in plants: the shikimate or chorismate and the polyketide pathway [78]. With the ongoing exploration of the pharmacological activities of anthraquinones, an increasing number of anthraquinones have been reported. Related research identified 280 unigenes involved in the synthesis of anthraquinones in *Rubia cordifolia* using SMRT technology. The study found that the *HMGR*, *PMK*, *DXS*, and *MCT* genes may play significant regulatory roles in the synthesis of anthraquinones in *R. cordifolia* [79]. Combining genomics, transcriptomics, and metabolomics, the analyses revealed that the significant amplification of genes associated with the anthraquinone pathway in *Rheum officinale*, along with the increase in gene quantity due to tetraploidy, may account for the high anthraquinone content in the *R. officinale* [80]. In the future, by continuously exploring more structural and regulatory genes related to anthraquinone synthesis using advanced technologies such as multi-omics, deeper analyses of the anthraquinone biosynthesis pathway and its regulatory mechanisms are expected.

### 2.3. The Impact of Environmental Factors on Secondary Metabolites

Generally, medicinal plants with excellent quality and unique efficacy from a specific place of origin are called authentic medicinal materials, and their regionality is one of the most remarkable characteristics [81]. The formation of genuine regional medicinal materials is closely related to their specific growth environments, including environmental factors such as temperature, light, mineral elements, and soil microorganisms. These factors affect the composition and content of secondary metabolites and jointly determine the quality and curative effect of medicinal plants, enabling the medicinal materials to exhibit higher medicinal value and better therapeutic results.

Climate change exerts an impact on the production of plant secondary metabolites. In the case of *Rubia cordifolia* L., its quality hinges on two secondary metabolites present in its roots, namely purpurin and mollugin. The research discovered that the contents of these two metabolites exhibit a positive correlation with latitude and longitude. Moreover, it was revealed that higher temperatures and shorter daylight durations are favorable for their synthesis [82]. The natural environment exerts a significant influence on the regional distribution patterns of soil microorganisms. Meanwhile, an intimate correlation exists between soil microorganisms and the secondary metabolism of plants. The quality of the genuine regional medicinal material ‘Guangchenpi’ varies remarkably due to different cultivation regions. By utilizing plant transcriptomics, metabolomics, and metagenomics, recent, previous studies have revealed that the high salinity and salt-tolerant microorganisms in the soil environment where Citri Reticulatae Pericarpium is cultivated in Xinhui, Guangdong Province significantly boost the synthesis of monoterpene content in the peel. This offers a theoretical breakthrough for analyzing the formation mechanism of the genuineness of ‘Guangchenpi’ [83]. Microorganisms capable of dissolving minerals play a crucial role in providing essential nutrients such as phosphorus (P), potassium (K), zinc (Zn), and selenium (Se), which are vital for plant growth and development [84]. These mineral elements directly influence the yield and quality of authentic medicinal materials. Recent studies have proved that exogenous selenium concentrations ranging from 50 to 100 mg/L can enhance photosynthesis and the absorption of mineral elements in tea trees, leading to an increase in their biomass. The levels of total selenium and organic selenium in tea have significantly risen, promoting the accumulation of tea polyphenols, theanine, flavonoids, and volatile secondary metabolites, thereby improving the nutritional quality of tea [85].

The special qualities of genuine regional medicinal materials result from the combined effects of their genotypes, specific ecological environments, and cultivation measures. Environmental factors can influence intracellular signal transduction pathways, alter the activity, synthesis, or degradation of transcription factors, and subsequently affect the transcription level of genes, regulate the expression of downstream genes, and thus impact effective substances such as secondary metabolites. Therefore, in-depth research on medicinal plants at the molecular level is of crucial significance for revealing their genuineness, and achieving this goal is inseparable from the support of genomics technologies.

## 3. Omics Application in Plant Secondary Metabolites

### 3.1. Genomics

The advancement of plant genomics has facilitated extensive investigations into the medicinal properties of plants by utilizing their genetic resources, as well as enhancing the biosynthesis of secondary metabolites [86,87]. A comprehensive examination of the genomes of medicinal plants enables the elucidation of their genetic foundations, the identification of functional genes, and the exploration of their associations with therapeutic efficacy. Genome sequences from the nucleus, chloroplast, and mitochondria provide a foundation for understanding plant genetic and evolution characteristics. For example, the chromosome-level genome of safflower has been de novo assembled and annotated to reveal the biosynthetic mechanisms of flavonoids and fatty acids in flowers and seeds, respectively [88]. Also, both the mitochondrial and chloroplast genomes of safflower have been sequenced [89,90], which not only enrich its genetic information, but also help to understand the phylogeny of safflower in the Compositae family. In addition, genomic technologies can identify key transcription factors and their target genes involved in metabolite biosynthesis, as well as cis-acting elements such as promoters, enhancers, and silencers, and construct transcriptional regulatory networks. Therefore, the genomic data contribute to a deeper understanding of regulatory and evolutionary mechanisms of specific metabolite biosynthesis.

In recent years, genomics has made significant advancements in medicinal plants (Figure 3). With the rapid development of sequencing technologies, long-read sequencing technologies (such as PacBio and ONT) and Hi-C sequencing technologies have shown great potential in improving genome assembly quality and resolving complex structural variations, making telomere-to-telomere (T2T) genome and pan-genome hot topics [91]. T2T genome assembly has been completed in an increasing number of medicinal plants, such as *Gynostemma pentaphyllum* [92], *Astragalus* [93], and *Myrtus communis* [94], providing a foundation for analyzing complex structural variations, gene functions, and plant breeding and improvement. The pan-genome encompasses the diversity of all DNA sequences in a species, offering a more comprehensive and accurate representation than the traditional single linear reference genome. Currently, pan-genome research has been conducted on crops such as soybean, rice, corn, and potato [95]; however, there has been limited investigation into medicinal plants. Additionally, research in related fields such as population genomics and evolutionary genomics have further promoted the study of available resources in medicinal plants. For example, a study conducted whole-genome association analysis with rutin content by re-sequencing 572 buckwheat samples, discovering *Fh06G015130* and *Fh03g007120*, which are speculated to play important roles in the biosynthesis of rutin [96].

Gene duplication, recombination, and fusion events can promote the evolution of specialized metabolites by altering gene functions or regulating metabolic networks. For instance, through collinearity analysis and functional annotation of the *P. somniferum* genome, researchers identified gene clusters related to the synthesis of morphine alkaloids [97]. The existence of gene clusters is not only related to the synthesis of specific metabolites but also may assist plants in adapting to particular ecological environments. One study conducted a comparative genomics analysis and discovered that the biosynthetic gene clusters responsible for momilactone in various plants were independently formed through convergent evolution, and all of these clusters exhibited similar defensive functions [98]. The production of new genes typically occurs through gene duplication, horizontal gene transfer, or gene rearrangement. These newly formed genes may play a role in the synthesis of secondary metabolites. For example, the tandem duplication of the *TPS* gene regulates the content of terpenoids in *Curcuma* plants [99]. Long terminal repeat (LTR) insertion plays a significant role in the evolution of plant genomes. The Tc1 scopia transposon in blood orange is inserted into the upstream regulatory region of the MYB transcription factor, which regulates anthocyanin biosynthesis by enhancing promoter activity [100].

In general, by analyzing the complete genome sequences of medicinal plants, genomics can reveal the biosynthetic pathways of metabolites and their regulatory mechanisms. This analysis aids in understanding the reasons behind the high expression of certain metabolites in specific plants, including the expansion of gene families due to gene duplication and evolutionary processes, as well as the regulatory differences among transcription factors and epigenetic modifications. Genomics serves as a crucial tool for uncovering the diversity and synthesis mechanisms of metabolites in medicinal plants, thereby establishing a theoretical foundation for the development and utilization of these plants.

**Figure 3 ijms-26-03867-f003:**
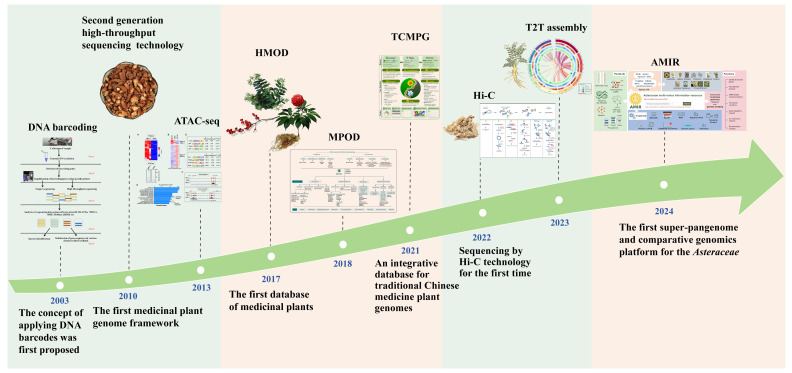
Development of genomic technologies for medicinal plants [93,101,102,103,104,105,106,107].

### 3.2. Metabolomics

Metabolomics focuses on the study of all endogenous small molecular metabolites with a relative molecular mass of less than 1000 in organisms. By detecting, identifying, and quantifying metabolites using advanced quantitative techniques, the metabolic characteristics and response mechanisms of organisms can be elucidated, thus providing a scientific basis for metabolic regulation and the improvement of metabolite content. For example, one study employed a targeted metabolomics analysis method based on ultraperformance liquid chromatography–high resolution mass spectrometry (UPLC-HRMS) to analyze mature lotus seeds. The study identified 767 metabolites and established multiple metabolic regulation maps, revealing that 18 DAP is the transition point when lotus seeds shift from active primary metabolism to massive deposition of secondary metabolites [108]. In the field of medicinal plant metabolomics, instrumental detection and analysis is one of the key technologies. The commonly used technical platforms include nuclear magnetic resonance (NMR), liquid chromatography–mass spectrometry (LC-MS), and gas chromatography–mass spectrometry (GC-MS), which can comprehensively and accurately analyze the composition and concentration of metabolites in medicinal plants. As research progresses, several metabonomic databases, such as KEGG [109], PMN [110], MPOD [103], PCMD [111], and KP337 [112], have been established. These databases encompass information on the metabolic networks of medicinal plants, chemical data on plant metabolites, and liquid chromatography–mass spectrometry datasets, thereby providing valuable resources for the study of plant metabolism. Traditional metabolic analysis methods, such as chromatography and spectrometry, typically focus on specific metabolites or metabolic pathways. This narrow focus makes it challenging to comprehensively cover all metabolites present in organisms, and the sensitivity of these methods is often insufficient for accurately detecting metabolites. As a result, they may fail to reflect the overall metabolic state of biological systems. Furthermore, traditional metabolic analysis requires extensive sample pretreatment, and metabolites can be significantly influenced by environmental factors such as temperature, drought, light [113], and hormones [114]. In contrast, metabolomics, which is based on genomic information that remains relatively stable throughout the life cycle of organisms, offers greater data support and can identify metabolites that traditional methods may not detect accurately. However, traditional metabolomics also entails sample homogenization, metabolite extraction, and mass spectrometry analysis. Such treatment neglects the spatial distribution information of metabolites within the different tissue structures of samples. In contrast, spatial metabolomics, as an emergent molecular imaging technology, integrates mass spectrometry imaging (MSI) and metabolomics techniques. It is capable of dissecting the distribution of metabolites across diverse tissues and organs in three dimensions: qualitatively, quantitatively, and spatially. This enables the acquisition of both the types and quantities of metabolites in distinct regions, as well as the revelation of the biological functions of various metabolites within their spatial configurations.

DL represents a novel research direction within the fields of AI. It enables the learning and abstraction of data features in a hierarchical manner by constructing multi-layer neural network structures, which gradually transform ‘low-level’ features into ‘high-level’ features, thereby facilitating the completion of complex learning tasks using relatively simple models. The application of DL can address the bottlenecks in metabolomics data collection, processing, metabolite identification, and the discovery of metabolic phenotypes and biomarkers. At present, several DL tools have been developed for data collection, processing, and downstream analysis in the field of metabolomics. These tools are utilized in collaborative tandem mass spectrometry (MS/MS) to identify and annotate “unknown” metabolites, such as DeepMass and MetDNA3, and to predict the compound classes from a fragmentation spectra, as seen in CANOPUS. Furthermore, when traditional methods, such as partial least squares (PLS), fail to extract the metabolic spectrum and biologically significant information, DL can effectively capture the metabolic characteristics of complex traits [115].

### 3.3. Integration of Multi-Omics

Multi-omics, as a novel systematic approach and technology for studying biology, can integrate various individual omics such as genomics, transcriptomics, proteomics, and metabolomics in an unbiased manner (Figure 4). Transcriptomics, which mainly studies gene expression at the RNA level, can help infer the functions of unannotated genes, making it possible to investigate the differences in gene expression among different cells, tissues, organs, or organisms [116]. Furthermore, single cell transcriptomics and spatial transcriptomics derived from transcriptomics will provide higher resolution for gene expression at a cell level. For example, a study used single cell transcriptomics to explore the spatial organization of MIA metabolism in *C. roseus* leaves, located 20 transcripts of MIA genes for the first time, and updated the MIA biosynthesis model [117]. The combination of scRNA-seq and spatial transcriptomics can further enhance the understanding of cell heterogeneity and tissue function. The development of the Spotiphy tool enables single-cell-resolution whole-transcriptome imaging. By integrating single-cell RNA sequencing data, spatial transcriptome data, and histological images, cell types and their spatial distribution can be more accurately identified [118]. Protein omics investigates the composition and activity of proteins at the cellular, tissue, or organismal level. It identifies proteins with unknown functions and monitors changes in the expression of key proteins under various conditions, with particular emphasis on the levels of related enzymes and regulatory factors involved in the metabolic processes of medicinal plants [119,120].

Integrative multi-omics analysis provides a systematic insights into the mechanisms and phenotypes of complex biological systems, helping to reveal the overall properties of biological systems, decipher the metabolic networks and regulatory mechanisms involved in plant evolution, as well as the exploration of the biosynthetic pathways of metabolites [121]. With the continuous optimization and increasing maturity of diverse technologies, the application scope of omics has been expanding. Recent research has integrated genomics, RNA sequencing, and metabolomics to clarify the biosynthetic pathway of leonurine, identifying critical enzymes involved in this pathway, including arginine decarboxylase (ADC) and uridine diphosphate (UDP)- glycosyltransferase (UGT) [122]. Another study compiled data on the metabolites and transcripts from the licorice rhizosphere, along with the associated rhizosphere microbiome, thereby elucidating the biosynthetic pathways of glycyrrhizin and glycyrrhizic acid. Their study established a close relationship between the bacterium *Bacillus* and the CYP72A154 enzyme necessary for the biosynthesis of glycyrrhizic acid, underscoring the critical role of the rhizosphere microbial community structure in the accumulation of glycyrrhizin [123].

The discovery of metabolite biosynthetic pathways in medicinal plants is significantly accelerated by multi-omics methods; however, these high-dimensional data sets are often underutilized. AI technologies offer revolutionary potential for the integration and analysis of multidisciplinary data [124]. By facilitating data integration, feature selection, pattern recognition, predictive modeling, and network analysis, AI can greatly enhance our ability to understand and interpret complex data sets. For instance, techniques such as LASSO, ridge regression, random forests, and DL can be used for feature selection or extraction, which aids in reducing dimensionality and identifying the characteristics of key genes, proteins, metabolites, or other predicted target phenotypes. Additionally, network-based methods can be utilized to infer gene regulatory networks, protein–protein interaction networks, or metabolic networks [125].

**Figure 4 ijms-26-03867-f004:**
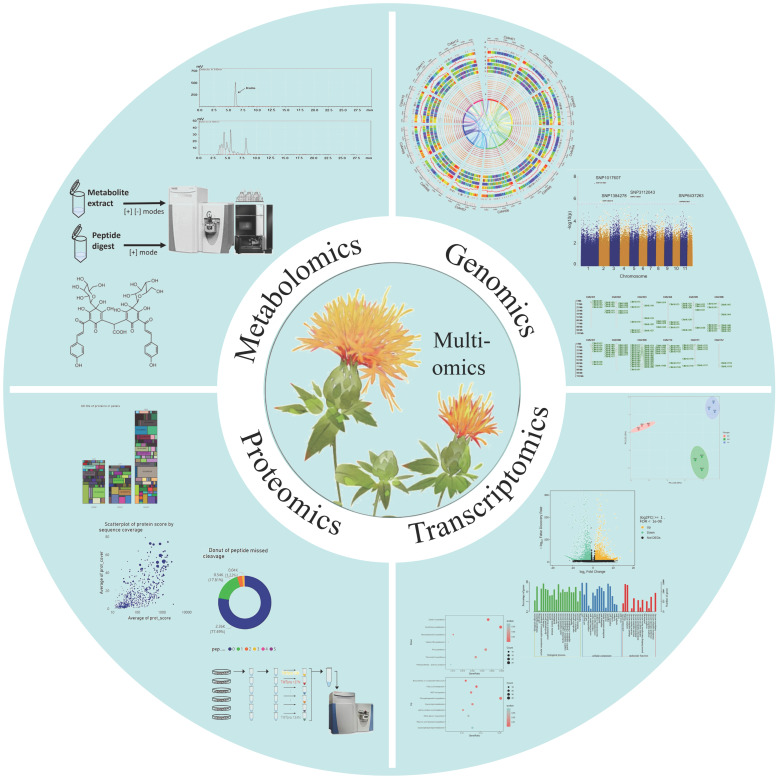
Overview of multi-omics technology [88,126,127,128,129,130,131].

## 4. A Case Study of Multi-Omics Application in Safflower Flavonoids

The application of multi-omics technologies in the study of safflower flavonoids not only helps to analyze the biosynthetic pathways of safflower flavonoids but also provides an important molecular basis for the molecular breeding and quality improvement of safflower (Figure 5). The construction of a high-quality safflower genome is an important prerequisite for understanding the genomic mechanism of safflower flavonoid synthesis. Compared to previous fragmented genome of safflower [132], one study has de novo assembled the first high-quality chromosome level genome based on the combination of PacBio Sequel sequencing and genetic linkage map. Further phylogenomics and multi-omics integrative analysis revealed that the tandem duplication event of chalcone synthase genes (*CHSs*) and their alternative splicing events may be essential for HSYA biosynthesis in safflower [88]. Integrative analysis of metabolomics and transcriptomics in flower developmental stages revealed that the key genes of flavonoid biosynthesis pathway, such as *4CLs*, *DFRs* and *ANRs*, were upregulated, while *CHIs*, *F3Hs* and *FLSs* were downregulated, indicating that metabolic flux may have a potential regulatory feedback mechanism or shift to specific flavonoid branches [133]. Based on a comparative transcriptome analysis, the distribution and collinear relationships of genes involved in the flavonoid biosynthesis pathway were examined. The results showed that the upstream genes such as *PALs*, *C4Hs*, and *4CLs* were highly expressed in stems and bracts, while *CHSs*, *CHIs*, and *F3Hs* were highly expressed in flowers, which was consistent with the tissue-specific distribution of flavonoids, indicating that the precursors of flavonoids might be synthesized in stems or leaves first and then transferred to flowers [134]. Multi-omics analysis also revealed that hormone MeJA can stimulate the synthesis of flavonoids in safflower, predominantly quinone chalcones, especially HSYA, yet its molecular mechanism remains to be further elucidated. Integrated analyses of metabolomics and transcriptomics technologies suggest that MeJA may upregulate the expression of upstream genes (such as *CHSs*, *CHIs*, and *HCTs*) in the flavonoid biosynthesis pathway and downregulate the expression of downstream genes (such as *F3Ms*, *ANRs*, and *ANSs*), thereby facilitating the biosynthesis of quinone chalcones [135], which is consistent with previous studies [136,137]. According to the expression patterns of *P450* and *UGT* genes in different development stages, light conditions, tissues and MeJA treatment, *OGT1* was identified as a highly active glycosyltransferase gene. Interestingly, the study found that MeJA inhibited the expression of *CtUGT3, CtUGT16* and *CtUGT25* [138,139], but the regulation mode of C-glucosides and O-glucosides biosynthesis has not been clarified.

UGTs can catalyze the formation of O-, N-, S and C-glycosides by using nucleotide activated sugars as glycosyl donors [140]. In 2014, a novel glucosyltransferase UGT73AE1 was first reported to have been cloned from safflower [141]. Subsequent studies have demonstrated that multiple *CtUGT* genes play multiple roles in the flavonoid biosynthesis pathway and color formation of safflower [139,142]. However, although *UGTs* have been identified as key genes for HSYA biosynthesis in safflower, few genes have been successfully cloned, and the specific biosynthesis pathway is unknown (Figure 5). Comprehensive multi-omics strategies were used to identify the UGTs used in flavonoid glycoside biosynthesis in safflower, and the 11 novel UGTs identified could catalyze the conversion of naringin chalcone and phloretin into corresponding O-glycosides, and their biochemical properties were analyzed based on AutoDock Vina (version 4.2) and enzyme kinetic analysis [134]. Further functional verification showed that the transient expression of *CtUGT3* in safflower protoplast resulted in a significant increase in the content of astragalus, which showed the glycosylation activities of 3-OH and 7-OH flavonoids in vitro. Meanwhile, Alphafold and AutoDock Vina (version 1.2.5) were used for molecular modeling and site-specific mutagenicity discovery. G15, T136, S276, and E384 are key catalytic residues in the glycosylation capacity of *CtUGT3* [143]. These studies reveal the importance of *UGT* genes in the biosynthesis of natural products in plants. At present, there are Genome annotation flow (GMind) and plant glycosyltransferase databases for mining UGTs. At the same time, relevant network tools can perform virtual screening of glycosyltransferase and predict the sugar donor of unknown glycosyltransferase, which provides resources for further research on UGTs [144]. In protein structure prediction, Alphafold 3 has significantly improved its accuracy compared with Alphafold 2, and its accuracy in predicting the interactions between proteins and ligands and proteins and nucleic acids has also been greatly improved. In addition, the combined structure of complexes including nucleic acids, small molecules, ions, and modified residues can also be predicted [145]. Resolving complex interactions between proteins and proteins or proteins and other molecules by AI provides new methods for mining potential *UGT* genes, and provides new insights for providing drug targets, protein modification, synthetic drugs, and synthetic biology.

In addition, ions not only maintain ion balance and electrophysiological functions within cells, but also influence metabolic pathways and secondary metabolite synthesis through complex signal transduction networks. However, there are few studies on ions in safflower, and most of them focus on the field of salt stress. Salt stress leads to a significant increase in the concentration of Na^+^ and a decrease in Ca^2+^, K^+^/Na^+^ ratio in safflower [146]. This change in the ionic balance not only affects the process of cellular osmotic regulation but also interferes with enzyme activity and gene expression, thereby affecting the metabolic process of flavonoids. Besides, the foliar application of silicon (Si) increased the contents of K^+^, Ca^2+^ and Si, while decreasing the absorption of Na^+^. The increase of ion content indicates that the selectivity of safflower for ion uptake improved following silicon application. This method enhances the activity of related enzymes by regulating the ion balance in cells, thus improving the tolerance of safflower under salt stress [147], providing a new perspective for understanding the resistance mechanism of plants. However, all these studies take the traditional approach of elemental analysis, which tends to focus on specific elements (e.g., K^+^, Ca^2+^, etc.) and lacks a systematic study of the overall ionic composition and interactions in the plant. Ionomics emphasizes the comprehensive analysis of all ions within the plant body. It employs modern high-throughput elemental analysis methods, such as inductively coupled plasma mass spectrometry (ICP-MS), inductively coupled plasma optical emission spectrometry (ICP-OES), X-ray fluorescence analysis, and neutron activation analysis, to simultaneously and quantitatively analyze multiple elements in plants [148]. However, there are few studies focused on the ionomics of medicinal plants.

**Figure 5 ijms-26-03867-f005:**
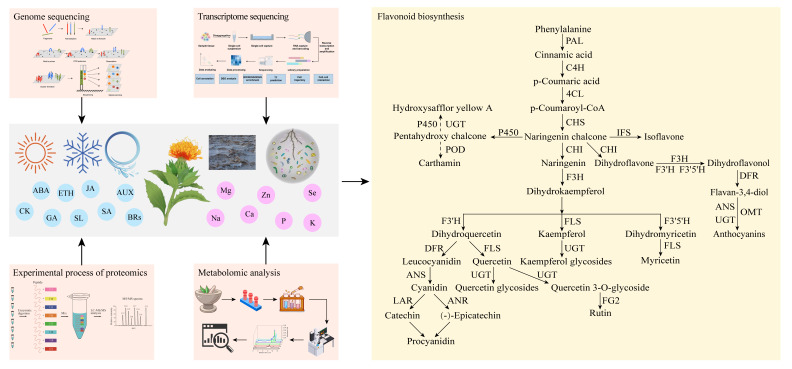
Regulation mechanism of secondary metabolites in safflower [149]. The red-shaded box represents the methods of omics technologies, the grey-shaded box represents the impact of environmental factors on safflower, and the yellow-shaded box represents the biosynthetic pathway of flavonoids in safflower. Note: The enzyme names are abbreviated as follows: PAL, phenylalanine ammonia lyase; C4H, cinnamic acid 4-hydroxylase; 4CL, 4-coumarate: CoA ligase; CHS, chalcone synthase; UGT, UDP-glycosyltransferase; P450, cytochrome P450; IFS, isoflavone synthase; POD, peroxidase; CHI, chalcone isomerase; F3H, flavanone 3-hydroxylase; F3′H, flavanone 3′-hydroxylase; F3′5′H, flavanone 3′,5′-hydroxylase; DFR, dihydroflavonol 4-reductase; ANS, anthocyanidin synthase; OMT, O-methyl transferases; FLS, flavonol synthase; LAR, leucoanthocyanidin reductase; ANR, anthocyanidin reductase; FG2, flavonol-3-O-glucoside L-rhamnosyltransferase.

## 5. Summary and Prospect

At present, the genomic quality of medicinal plants is inconsistent due to variations in annotation and sequencing technologies, which significantly hampers the identification of secondary metabolic gene clusters and functional genes. A high-quality genome serves as a crucial foundation for further research on the biosynthesis and regulation of active components in medicinal plants. The assembly of T2T genome can provide a gapless genome sequence, ensuring that the key genes involved in the biosynthesis pathways of secondary metabolites are fully annotated. Pan-genomes are crucial in breeding practices and plant domestication, facilitating the discovery of trait-associated loci and identifying variable genes for editing in plant genome editing and in establishing new editing systems. However, only 11 medicinal plant genomes have reached the T2T genome level [150], and the lack of standardized processes for pan-genomic studies hinders the comparison and validation of analysis results across different studies [151]. With the decreasing costs of sequencing and the ongoing advancements in genome annotation methods, it is anticipated that an increasing number of medicinal plants will achieve complete T2T genome assemblies. The integration of pan-genome with this progress establishes a robust genetic basis for gene mining utilizing high-throughput genomic data, thereby advancing the research of medicinal plants to new heights.

In the workflow of metabolomics analysis, innovations in spatial metabolomics, single cell metabolomics, MS technology and data acquisition methods have developed rapidly in recent years. However, no single detection technology can identify all the metabolites present in a sample [152]. Moreover, there are numerous types of metabolites and great differences in their concentrations. Therefore, it is imperative to advance the methodological standardization of metabolomics, encompassing the entire process from sample collection, preparation, and processing to data analysis and interpretation [153]. This standardization aims to enhance the accuracy and precision in detecting subtle differences in metabolic phenotypes. Along with technology matures, the experimental processes and data analysis in metabolomics will be more standardized and automated. Enhancements in sample processing, data handling, and analytical methods will substantially improve the repeatability and reliability of research findings.

The integration and application of novel omics methods offer a fresh perspective for studying secondary metabolic pathways through a multi-omics approach. By integrating ATAC-seq, Hi-C, and DAP-seq, multi-omics analysis can comprehensively analysis the regulatory network of genes that encode key enzymes in secondary metabolic pathways from multiple levels such as chromatin accessibility, three-dimensional genome structure, and transcription factor binding sites [154,155,156]. Furthermore, in recent years, spatial omics technology has made remarkable progress in the study of plant secondary metabolites. By combining scRNA-seq and spatial metabolomics, it can analyze the biosynthetic pathway, cell specificity and response mechanism of plant secondary metabolites to the environment. In related studies, the integrated application of spatial metabolomics such as mass spectrometry imaging (MSI) and matrix-assisted laser desorption/ionization (MALDI)-MSI with single-cell transcriptome sequencing and open chromatin sequencing (ATAC-seq) in taxus revealed new key enzyme genes, regulatory factors, and transporters in the paclitaxel synthesis pathway. A novel cell-level map of the paclitaxel synthesis pathway was mapped [157]. With advances in spatial sequencing, imaging techniques, and bioinformatics analysis, spatial multi-omics is expected to reveal the mechanisms of synthesis, transport, and accumulation of natural products in medicinal plants. However, it is worth noting that the integration of omics data is still a challenge, and the stability of various omics data collection and the characteristics of different omics data sets need to be fully considered. Therefore, it is particularly important to integrate data and mathematical models. In the future, more complete databases and more advanced analytical tools are needed to obtain reliable data with wider coverage [152].

AI technology has emerged as a crucial tool for predicting enzyme structures in medicinal plant research. A study based on virtual screening of predicted protein structures efficiently identified novel glycosyltransferases involved in salidroside biosynthesis. The experimental results show that Ach15909 exhibits high catalytic activity [158]. This method offers significant advantages for mining secondary metabolic synthetases in non-model plants, particularly when working with large datasets that include thousands of candidate genes. It can effectively enhance screening efficiency and establish a new paradigm for plant enzyme discovery. On the other hand, the rapid development of artificial intelligence has promoted the development and utilization of various modeling tools. A study based on the known crystal structure of terpene synthases, researchers conducted FgMS homology modeling to analyze non-conserved amino acid sites near non-active pockets of protein sequences [159]. This approach has significantly enhanced the understanding of the synthesis pathways of secondary metabolites. In addition, AI technology serves as a powerful tool for genome annotation, offering broad application prospects. For instance, Convolutional Neural Networks (CNNs) are among the DL models with significant potential applications in genome annotation. However, the impact of AI technology on genome annotation has not yet transformed the field as it has in protein structure prediction [160]. Nevertheless, with the ongoing development of new models and the rapid expansion of omics data, AI is anticipated to promote a major breakthrough in genome annotation.

In light of the ongoing advancements in genomic sequencing and AI technologies, we believe that integration of AI-based genomics and multi-omics analysis will assume an increasingly significant role in the study and application of medicinal plants. Through the continuous optimization and refinement of efficient algorithms, it is expected that we will reveal the mysteries behind the complex metabolic pathways, underlying the bioactive constituents of medicinal plants, identify potential functional genes, and offer more precise and efficient guidance for the breeding, cultivation, development, and utilization of medicinal plants.

## Figures and Tables

**Figure 1 ijms-26-03867-f001:**
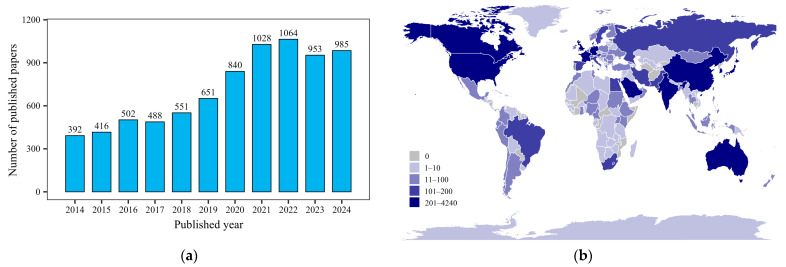
Number of articles published on genomic research of medicinal plants over the past 10 years. (**a**) Statistics of the number of publications by year. (**b**) Statistics of the number of publications by country. The visualization was generated using R (version 4.4.1) with the ggplot2 (version 3.5.1), dplyr (version 1.1.4), and cowplot (version 1.1.3) packages.

**Figure 2 ijms-26-03867-f002:**
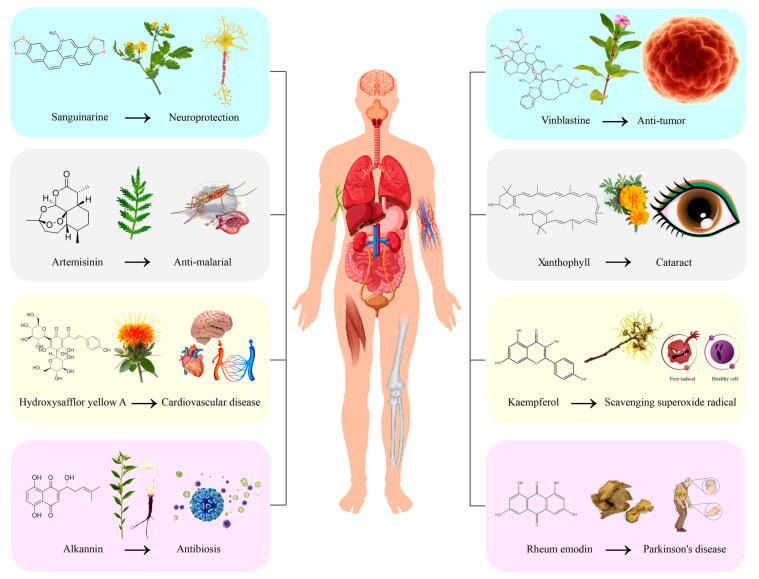
Common secondary metabolites and their medicinal values. The blue-shaded box represents alkaloids, the gray-shaded box represents terpenoids, the yellow-shaded box represents flavonoids, and the pink-shaded box represents quinones. The figure was created using Adobe Illustrator 2022.

**Table 1 ijms-26-03867-t001:** Common secondary metabolites and their medicinal value.

Secondary Metabolites		Plant Sources	Pharmacological Activity	References
Alkaloid	Vincristine	*Catharanthus roseus*	Anti-tumor	[10]
	Camptothecin	*Camptotheca acuminata* Decne.	Anti-tumor, treatment of psoriasis	[11,12]
	Piperine	*Piper nigrum* L. *Piper longum* L.	Reduce insulin resistance, anti-inflammatory, anti-liver steatosis, improve bioavailability	[13]
	Morphine	*Papaver somniferum* L.	Pain relief, cough suppression, treatment of cardiovascular diseases	[14,15]
	Papaverine	*Papaver somniferum* L.	Anti-tumor, analgesic, treatment of cardiovascular diseases	[16,17]
	Tetrandrine	*Stephania tetrandra* S. Moore	Anti-inflammatory, cardiovascular diseases, treatment of silicosis	[18,19]
	Bloodroot alkaloid	*Chelidonium majus* *Corydalis edulis* Maxim. *Macleaya cordata* (Willd.) R. Br.	Anti-tumor, antibacterial, anti-osteoporosis, neuroprotection	[20,21]
	Berberine	*Coptis chinensis* Franch *Phellodendri Cortex* *Berberidis Radix*	Anti-tumor, treatment of cardiovascular diseases and diabetes	[22,23]
Terpenoids	Artemisinin	*Artemisia car* *vifolia*	Anti-malarial, anti-tumor, treatment of cardiovascular diseases and polycystic ovary syndrome	[24,25]
	Paclitaxel	*Taxus chinensis*	Anti-tumor	[26]
	Tanshinon	*Salvia miltiorrhiza* Bunge	Antibacterial, anti-inflammatory, treatment of cardiovascular diseases	[27,28]
	Ginsenoside	*Panax ginseng* C. A. Mey.	Anti-tumor, anti-inflammatory, anti-allergic reaction, antidepressant	[29,30]
	Menthol	*Mentha canadensis* L.	Anti-tumor, antibacterial, analgesic	[31,32]
	Camphor	*Camphora officinarum* Nees ex Wall.	Anti-tumor, antibacterial, analgesic	[31,32]
	Thymol	*Tachyspermum ammi* *Origanum vulgare* L.	Anti-tumor, anti-inflammatory, neuroprotective	[33,34]
	Lutein	*Tagetes erecta* L. *Calendula officinalis* L. *Brassica oleracea* var. *capitata Linnaeus*	Treatment of neurodegenerative diseases and eye diseases	[35,36]
Phenols	HSYA	*Carthamus tinctorius* L.	Anti-tumor, neuroprotection, treatment of cardiovascular diseases	[9,37,38]
	Baicalein	*Scutellaria baicalensis* Georgi *Oroxylum indicum* *Plantago major* L.	Anti-tumor, antibacterial, antiviral, treatment of cardiovascular diseases	[39,40,41,42]
	Quercetin	*Flos Sophorae Immaturus* *Notoginseng Radix* *Ginkgo biloba* L.	Anti-tumor, antibacterial, antiviral, treatment of cardiovascular diseases	[43,44]
	Kaempferol	*Kaempferia galanga* L. *Forsythia suspensa* (Thunb.) Vahl *Ginkgo biloba* L.	Anti-tumor, antibacterial, anti-inflammatory, diabetes, treatment of cardiovascular diseases	[45,46]
	Anthocyanidin	*Vaccinium* spp. *Lycium ruthenicum* *Brassica oleracea* var. *capitata Linnaeus*	Anti-tumor, treatment of cardiovascular diseases, vision protection	[47,48]
	Luteolin	*Dracocephalum integrifolium* Bge. *Lonicera japonica* Thunb. *Perilla frutescens* (L.) Britt.	Anti-tumor, anti-inflammatory, antioxidant	[49,50]
	Genistein	*Sophora japonica* Linn *Euchresta japonica* Benth. ex Oliv.	Anti-tumor, anti-inflammatory, antibacterial, treatment of cardiovascular diseases	[51]
	Catechin	*Camellia sinensis* (L.) O. Ktze.	Anti-tumor, treatment of diabetes and cardiovascular diseases	[52,53]
Quinones	Aloe-emodin	*Cassia occidentalis* *Rheum palmatum* L. *Polygonum multiflorum* Thunb	Anti-tumor, antiviral, anti-inflammatory, immunological modulation	[54]
	Alkannin	*Lithospermum erythrorhizon* *Alkanna tinctoria* (L.)	Anti-tumor, wound healing, antibacterial	[55]
	Coenzyme Q10	Widely present in organisms	Antioxidant, improve heart health	[56]
	Oncocalyxone A	*Cordia oncocalyx*	Anti-inflammatory, analgesic, neuroinhibitory	[57]
	Emodin	*Rheum palmatum* (Chinese rhubarb)	Neurodegenerative diseases, Parkinson’s disease	[58]

## Data Availability

Not applicable.

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
