# Peer review of "Research Progress of Genomics Applications in Secondary Metabolites of Medicinal Plants: A Case Study in Safflower"

_ijms, 2025, doi:10.3390/ijms26083867_

Round 1

Reviewer 1 Report

Comments and Suggestions for Authors

This review work is not well focused or organized. It has an unnecessarily long introduction before reaching the central theme that is the case of the Safflower. It is understood that some context is needed to explain the potential of the omics approach to assess the mechanisms and targets of safflower components. However, this temporary technological evolution is well known and could be applied to any species without the need for such an exhaustive review in which, logically, hundreds of examples of its application can be given. In addition to the above, there are already quite a few works that highlight the use of multi-omics tools to analyze medicinal plants. It is only on page 11 of the manuscript that the "case of the Carthamus" is addressed. This section presents details of what is known about the species in terms of its metabolomics and some biological activities. What is displayed in Figure 5 is interesting since it includes not only elements of metabolomics, but also of ionobiology and ionomics, an aspect that is little addressed in many manuscripts. Unfortunately, in this work, although it is shown in Figure 5, nothing is explained about it either. In my opinion, although this species is very interesting for its pharmacognostic value, the authors should present this review in a more direct way, guiding them with respect to unresolved research flanks. If the regulation of the synthesis and some activities of HSYA are known, why not talk about the potential of nascent areas such as AI-driven drug discovery in the study of secondary metabolites of plants, and how this type of growing technology can help in the identification of targets, prediction of bioactivities, toxicity and synergy, or the optimization of the extraction process of target compounds.

Author Response

Comments 1:[ This review work is not well focused or organized. It has an unnecessarily long introduction before reaching the central theme that is the case of the Safflower. It is understood that some context is needed to explain the potential of the omics approach to assess the mechanisms and targets of safflower components. However, this temporary technological evolution is well known and could be applied to any species without the need for such an exhaustive review in which, logically, hundreds of examples of its application can be given.]

Response 1: Thanks for your comments. According to your suggestion, we have adjusted the length of the section on genomics methodology, merged the topics of protein genomics and transcriptomics into the content on multi-omics, and made appropriate deletions. We have retained the sections on genomics, metabolomics, and multi-omics, as these are closely related to the theme of this review. This change can be found – page 10, paragraph 3-4, and line 359-384; page 11, paragraph 2, and line 388-411; page 12, paragraph 1-3, and line 412-462.

Comments 2:[ This section presents details of what is known about the species in terms of its metabolomics and some biological activities. What is displayed in Figure 5 is interesting since it includes not only elements of metabolomics, but also of ionobiology and ionomics, an aspect that is little addressed in many manuscripts. Unfortunately, in this work, although it is shown in Figure 5, nothing is explained about it either. In my opinion, although this species is very interesting for its pharmacognostic value, the authors should present this review in a more direct way, guiding them with respect to unresolved research flanks.]

Response 2: Thanks for your comments. According to your suggestion, we have elaborated on the influence of mineral elements on the secondary metabolites of medicinal plants in section “2.3. The Impact of Environmental Factors on Secondary Metabolites” and added the existing research on mineral elements in safflower in section “4. A Case Study of Multi-Omics Application in Safflower Flavonoids”. This change can be found – page 9, paragraph 2, and line 303-312; page 15, paragraph 2, and line 554-575.

Comments 3:[ If the regulation of the synthesis and some activities of HSYA are known, why not talk about the potential of nascent areas such as AI-driven drug discovery in the study of secondary metabolites of plants, and how this type of growing technology can help in the identification of targets, prediction of bioactivities, toxicity and synergy, or the optimization of the extraction process of target compounds.]

Response 3: Thank you for your suggestion. Although progress has been made in the regulation of HSYA, the enzymes that directly catalyze its synthesis remain unclear. This paper aims to elucidate the synthesis mechanisms of secondary metabolites and to conduct functional gene mining from a botanical perspective. The field of artificial intelligence is indeed one of the current research hotspots. According to your suggestion, we complemented the AI technology in multi-omics analysis, metabolite mining, enzyme catalysis and structure prediction. This change can be found – page 12, paragraph 2, and line 428-441; page 13, paragraph 3, and line 478-488; page 14, paragraph 2, and line 531-553; page 17, paragraph 2, and line 637-656.

Reviewer 2 Report

Comments and Suggestions for Authors

The manuscript requires significant revision before it can be considered for publication.

1. I recommend that the authors include a new section that highlights the relationship between genomic research and the main categories of secondary metabolites, their subcategories, biosynthetic pathways, and evidence for metabolite identification or discovery.

2. Furthermore, suggestions to the authors, it is important to provide an additional section, in  which  a comprehensive discussion of  various studies on 13C-NMR and MS dereplication, as well as accessible databases that utilize contemporary metabolomic techniques, is provided. This section provides evidence of the reported compounds and outlines key strategies for future investigations into undiscovered metabolites.

 3. The authors may present comprehensive advantages of the genomic approach over traditional secondary metabolite screening methods.

4. The composition and Function of Plant Secondary Metabolites require a complete revision. Currently, its content covers only three groups: alkaloids, terpenoids, and flavonoids. To provide a more thorough and precise overview of the topic, the authors should broaden their examination to encompass all the major categories of secondary metabolites. Additionally, they should incorporate the primary metabolites found in plant metabolomes. This more extensive approach provides a more complete and accurate representation of the subject.

5. The genetic Regulation of Plant Secondary Metabolites requires complete revision. This segment should incorporate specific details that connect it to the preceding section, particularly regarding the initial stages of producing various main categories of secondary metabolites. Furthermore, there is an absence of information on the current state of research into the complete genomes of medicinal plants, which is essential for a comprehensive overview of secondary metabolites reported to date. The authors are asked to provide an in-depth discussion on this crucial aspect.

6. Environmental factors play crucial roles in the production of secondary metabolites. The authors are required to provide comprehensive details on how mineral elements influence the deregulation of secondary metabolite biosynthesis. To offer a more thorough and precise representation of this section, it is suggested that the authors incorporate evidence information on these elements using state-of-the-art techniques, such as ICP-MS.

7. The authors are requested to highlight the limitations of the current study.

Author Response

Comments 1:[ I recommend that the authors include a new section that highlights the relationship between genomic research and the main categories of secondary metabolites, their subcategories, biosynthetic pathways, and evidence for metabolite identification or discovery.]

Response 1: Thanks for your comments. According to your suggestion, we have added the related content in section “3.1. Genomics”, where we have elaborated on the relationship between genomics and secondary metabolite biosynthesis. The emergence of gene clusters, new genes, gene duplication, recombination, and fusion events, as well as the insertion of transposons, can all promote the evolution of specialized metabolites by altering regulatory networks, thereby enriching certain medicinal plants with specific secondary metabolites. Genomic research can delve deeply into these issues and provide important evidence for elucidating the mechanisms underlying the formation of secondary metabolites in medicinal plants. This change can be found – page 10, paragraph 3-4, and line 359-384.

Comments 2:[ Furthermore, suggestions to the authors, it is important to provide an additional section, in which a comprehensive discussion of various studies on 13C-NMR and MS dereplication, as well as accessible databases that utilize contemporary metabolomic techniques, is provided. This section provides evidence of the reported compounds and outlines key strategies for future investigations into undiscovered metabolites.]

Response 2: Thank you for your suggestion. We fully agree with your suggestion to add more content related to technologies and databases. To this end, in section “3.2. Metabolomics”, we have elaborated on the commonly used technological platforms, established databases, and the advantages of genomics-based metabolomics compared to traditional metabolic analysis methods. We have particularly emphasized the high-throughput, high-sensitivity, and high-accuracy capabilities of metabolomics technologies in the identification and quantification of metabolites. The application of cutting-edge technologies such as spatial metabolomics is also expounded.  These strengths make metabolomics a powerful tool for studying changes in metabolites within complex biological systems. Additionally, in the “Summary and Prospect” section, we have expanded the discussion with challenges faced by metabolomics and provided our outlook for the future. This change can be found – page 11, paragraph 2, and line 388-427; page 16, paragraph 2, and line 603-614.

Comments 3:[ The authors may present comprehensive advantages of the genomic approach over traditional secondary metabolite screening methods.]

Response 3: Thanks for your comments. Regarding this part, we have provided a response in Response 2. Specifically, traditional metabolic analysis methods focus on specific metabolites or pathways, with low sensitivity and extensive sample pre-treatment required, making it difficult to reflect the overall metabolic state of a biological system. In contrast, metabolomics, based on the relatively stable genomic information of an organism, provides comprehensive data support and can identify metabolites that traditional methods fail to detect. This change can be found – page 11, paragraph 2, and line 408-418.

Comments 4:[ The composition and Function of Plant Secondary Metabolites require a complete revision. Currently, its content covers only three groups: alkaloids, terpenoids, and flavonoids. To provide a more thorough and precise overview of the topic, the authors should broaden their examination to encompass all the major categories of secondary metabolites. Additionally, they should incorporate the primary metabolites found in plant metabolomes. This more extensive approach provides a more complete and accurate representation of the subject.]

Response 4: Thanks for your comments. To provide a more comprehensive and accurate overview of the topic, we have revised the classification and functions of plant secondary metabolites. In addition to the originally included alkaloids, terpenoids, and flavonoids, we have added a detailed description of quinones. Besides, due to the large number of primary and secondary metabolites in plants, and the fact that primary metabolites are often related to plant growth and development, other factors were not considered. This ensures that the article remains concise and focused. This change can be found – page 4, paragraph 5, and line 174-191.

Comments 5:[ The genetic Regulation of Plant Secondary Metabolites requires complete revision. This segment should incorporate specific details that connect it to the preceding section, particularly regarding the initial stages of producing various main categories of secondary metabolites. Furthermore, there is an absence of information on the current state of research into the complete genomes of medicinal plants, which is essential for a comprehensive overview of secondary metabolites reported to date. The authors are asked to provide an in-depth discussion on this crucial aspect.]

Response 5: Thanks for your comments. We have thoroughly revised the section of genetic regulation, and discussed the regulation of secondary metabolites mentioned in the previous section in turn. First, we began with the key enzymes in the biosynthetic pathways of secondary metabolites and elaborated on their roles in the synthesis stage. Second, we conducted an in-depth analysis of the regulatory mechanisms of gene expression from the perspective of transcriptional regulation. This is intended to help readers better understand the molecular mechanisms underlying the biosynthesis of secondary metabolites. Regarding the current status of complete genome research in medicinal plants, we have mentioned it in the introduction. This change can be found – page 5, paragraph 3, and line 200-232; page 6, paragraph 1-3, and line 233-273.

Comments 6:[ Environmental factors play crucial roles in the production of secondary metabolites. The authors are required to provide comprehensive details on how mineral elements influence the deregulation of secondary metabolite biosynthesis. To offer a more thorough and precise representation of this section, it is suggested that the authors incorporate evidence information on these elements using state-of-the-art techniques, such as ICP-MS.]

Response 6: Thanks for your comments. We fully agree with the important role of mineral elements in the biosynthesis of secondary metabolites and have revised the relevant content based on your suggestions. In section “2.3. The Impact of Environmental Factors on Secondary Metabolites”, we elaborated on the impact of mineral elements on secondary metabolites in medicinal plants. In section “4. A Case Study of Multi-Omics Application in Safflower Flavonoids”, we added research on mineral elements in safflower and discussed ionomics based on technologies such as ICP-MS and ICP-OES. These modern high-throughput elemental analysis methods can provide strong support for studying the relationship between mineral elements and secondary metabolites. This change can be found – page 9, paragraph 2, and line 303-312; page 15, paragraph 2, and line 554-575.

Comments 7:[ The authors are requested to highlight the limitations of the current study.]

Response 7: Thanks for your suggestion. This part has been expanded and mentioned in the Summary and Prospect section. From the perspective of genomics, differences in genome quality hinder the identification of secondary metabolite gene clusters and functional genes. The experimental processes and data analysis in metabolomics still need to be more standardized. Moreover, the integration of multi-omics data remains a challenge. This change can be found – page 16, paragraph 1, and line 595-602; page 16, paragraph 2, and line 605-614; page 17, paragraph 1, and line 631-636.

Round 2

Reviewer 1 Report

Comments and Suggestions for Authors

This manuscript, initially rejected by this reviewer, is now presented in a more coherent and focused manner. Most of the suggestions were accepted and skillfully incorporated in those places where the original document presented serious flaws. The changes and clarifications cause a significant improvement in the article and therefore I recommend considering this revised version for publication in IJMS.